# Effect of different running protocols on bone morphology and microarchitecture of the forelimbs in a male Wistar rat model

Andy Xavier[1,2,3]*, Céline Bourzac[1,4], Morad Bensidhoum[1], Catherine Mura[2], Hugues Portier[1☉], Stéphane Pallu[1,3☉]

1 Laboratoire B3OA UMR7052 CNRS U1271 INSERM, Université de Paris, Paris, France, 2 Laboratoire INEM UMR7355 CNRS, Université d'Orléans, Orléans, France, 3 Sport, Physical Activity, Rehabilitation and Movement for Performance and Health (SAPRéM), Université d'Orléans, Orléans, France, 4 Plateforme de Recherche Biomédicale, Ecole Nationale Vétérinaire d'Alfort, Maisons-Alfort, France

☉ These authors contributed equally to this work.
* andy.xavier@univ-orleans.fr

## Abstract

### Background

It is accepted that the metabolic response of bone tissue depends on the intensity of the mechanical loads, but also on the type and frequency of stress applied to it. Physical exercise such as running involves stresses which, under certain conditions, have been shown to have the best osteogenic effects. However, at high intensity, it can be deleterious for bone tissue. Consequently, there is no clear consensus as to which running modality would have the best osteogenic effects.

### Aim

Our objective was to compare the effects of three running modalities on morphological and micro-architectural parameters on forelimb bones.

### Methods

Forty male Wistar rats were randomly divided into four groups: high intensity interval training (HIIT), continuous running, combined running ((alternating HIIT and continuous modalities) and sedentary (control). The morphometry, trabecular microarchitecture and cortical porosity of the ulna, radius and humerus were analyzed using micro-tomography.

### Results

All three running modalities resulted in bone adaptation, with an increase in the diaphyseal diameter of all three bones. The combined running protocol had positive effects on the trabecular thickness in the distal ulna. The HIIT protocol resulted in an increase in both medio-lateral diameter and cortical bone area over total area (Ct.Ar/Tt.Ar) at the ulnar shaft compared with sedentary condition. Moreover, the HIIT protocol decreased the mean surface area of the medulla (Ma.Ar) according to sedentary condition at the ulnar shaft.

Data Availability Statement: All relevant data are within the paper and its Supporting Information files.

**Funding:** Gérond'if DIM Longévité & Vieillissement Région île de France. The funders had no role in study design, data collection and analysis, decision to publish, or preparation of the manuscript.

## Conclusion

This study has shown that HIIT resulted in a decrease in trabecular bone fraction in favor of cortical bone area at the ulna.

## Introduction

With the ageing of the population, degenerative diseases such as musculoskeletal disorders (osteoarthritis, osteoporosis, for instance) represent a major public health issue [1]. Given the increase in healthcare expenditure associated with these diseases, new therapies and prevention are being explored. Physical exercise is recognized for its beneficial effects on the whole body and particularly on the musculoskeletal system [2]. Muscular activity during exercise generates mechanical stress and releases myokines and cytokines which have beneficial effects on bone tissue [3]. Bone tissue is made up of two complementary microarchitectural (trabecular and cortical) structures. A such organisation adapts in response to mechanical stress. Consequently, depending on the type and the frequency of physical exercise applied. These two structures exhibit different metabolic answers [4]. According to Wolff's law [5], bone tissue models and remodels itself according to the stresses it undergoes. According to Frost's mechanostat theory [6], however, the osteogenic effects of physical exercise are dependent on a threshold intensity and a frequency of mechanical stress which remains to be defined [7]. This intensity is highly dependent on the type of activity (weight-bearing activity (running, jumping), loading with vibration (vertical whole vibration), partial unloading (cycling), or total unloading (swimming)). Indeed, Scheuren *et al.*, 2020 [8] highlight the mechanical signals influencing bone formation and resorption in the local environment *in vivo*. Consequently, the mechano-regulation of bone tissue logarithmically depends on the loading frequency. For frequencies below a certain threshold, the effects on bone tissue are catabolic, while higher frequencies have anabolic consequences on bone status.

Animal models remain necessary to improve the understanding of these mechanisms. Portier *et al.*, 2020, [9] have reviewed all the effects of physical exercise on bone tissue in rat-model studies. They concluded that interval running on a high-intensity treadmill (as well as repeated jumps at a certain frequency and height (jumps performed between 30 and 60 cm, 10 to 50 times a day, three to five days a week for eight weeks) induced the best osteogenic effects (increase in Bone Mineral Density (BMD), Bone Volume/Tissue Volume (BV/TV) and Trabecular number (Tb.N)) in male and female rats. However, the rat jumping model cannot strictly be applied as a therapy to humans, as it would represent a three-storey fall on a human scale, obviously leading to fractures.

Running can be performed in two main modalities: moderate continuous running and interval running. Continuous running consists of moderate exercises to constant intensity, performed over a prolonged period without interruption, lasting from 30 minutes to several hours. The intensity corresponds to around 50 to 80% of maximal oxygen uptake ($VO_2$max). High-Intensity Interval Training (HIIT) consists of alternating periods of high-intensity exercise and recovery (active or rest). Sessions are generally short, often lasting between 20 and 30 minutes. HIIT has a very high intensity during periods of effort, which can reach maximum $VO_2$max (90% to 110% $VO_2$max) [10–12]. Historically, moderate continuous running has been the most widely used modality in studies on bone tissue due to its ease of application [13–16]. In osteopenia conditions, compared to sedentariness, continuous running has shown osteoprotective effects [13, 16]. Very few experimental studies have previously compared the

effects of both moderate continuous and high interval running on the bone status [17–19]. The interval model seems to have a better osteogenic or osteoprotective effect. Nevertheless, in both humans and animals, high-intensity running cannot be practiced on a daily basis because it leads to a depletion of the organism. Indeed, Polisel *et al.*, 2021 [20], have shown that HIIT in healthy mice 26 week-old at the end of the protocol), trained for 30 minutes five times a week for 10 weeks, induced a decrease in bone strength). In order compare the effects of these running modalities on bone tissue and understand the osteogenic mechanisms involved, there is a need to quantify the total workload.

Few studies have used a protocol of moderate continuous running and interval running based on the overall mean exercise intensity, calculated according to the equation described by Mujika *et al.*, 1995, [17, 18]. Boudenot *et al.*, 2015 [19], reported that interval running improved the BMD by 20% at the femoral neck and over the whole body. There is a third running modality, which combines continuous running and interval running over several training sessions. The combined running is practiced by top-level athletes during their training since the 20th century [21]. However, scarce research has been achieved on the effects of this modality on bone tissue. The study by Wazzani *et al.*, 2023 [22], shed light these three running modalities and their respective effects on femoral properties. In the combined exercise group, the trabecular thickness (Tb.Th) in the proximal femur was higher compared to respective results in the other exercise groups (interval and continuous), and in the control sedentary condition (p<0.001). Furthermore, the trabecular BMD measured by micro-computed tomography (µCT) in the interval exercise group was higher compared to respective results in all other experimental groups.

The majority of studies on bone tissue in animal models report their results mainly in both femur and tibia and in the lumbar spine because of the deleterious effects of human diseases such as osteoporosis on these sites. Running has been shown to have positive effects on the hindlimbs in both humans and animal models. During locomotion in quadrupeds, the forelimbs are used to absorb impact and modify posture so that the hindlimbs can propel the body. It would therefore appear that the forelimbs absorb just as much mechanical stress as the hindlimbs [23]. Thus, in animal models, a bone adaptation to exercise for the forelimbs could be envisaged. To date, no study in rats has compared the impact of three different running modalities (HIIT, continuous, combined) on the forelimbs.

The aim of this study was to explore the effects of three treadmill running modalities (HIIT, continuous, combined), in a male Wistar rat model, on bone quality parameters in the forelimbs (humerus, radius, ulna) and determine which protocol is the most osteogenic. The study also aims to determine which specific anatomical regions of the forelimb bones are the most subjected to changes in their microarchitecture. These effects will be assessed by bone morphological analyses and µCT microarchitectural analysis.

## Material & methods

### 2.1. Animals

Forty 5-week-old, male, Wistar rats were randomly divided into four groups (4 rats per standard cage) and acclimatized for a week on a 12h/12h photoperiod cycle. The rats were obtained from Janvier Labs (Le genet-St-Isle, France) and fed ad libitum with a standard diet (Genestil, Royaucourt, France) containing 53.4% carbohydrate, 19.2% crude protein, 11.3% moisture, 6.1% crude cellulose, 5.9% crude ash and 4.1% fat, and ad libitum access to water. Two weeks after acclimatization, the rats were divided into four experimental groups of ten rats. The Sedentary control group (SED), the Continuous Running group (CR), the High

**Table 1. Rat weight and length, maximal aerobic speed (MAS) and BMI (body mass index) at T0 (beginning of the study), and T8 (end of the running protocol).**

| Parameters | Time | SED | HIIT | CR | ComR |
|---|---|---|---|---|---|
| **Weight (g)** | T0 | 199 ± 8 | 197 ± 7 | 192 ± 6 | 201 ± 9 |
| | T8 | 492 ± 36 | 475 ± 31 | 486 ± 33 | 491 ± 16 |
| **Length (cm)** | T0 | 19.8 ± 0.5 | 19.8 ± 0.4 | 19.7 ± 0.4 | 19.6 ± 0.5 |
| | T8 | 26.7 ± 1.1 | 26.4 ± 0.6 | 26.4 ± 0.6 | 26.5 ± 0.7 |
| **BMI (g/cm²)** | T0 | 0.51 ± 0.02 | 0.50 ± 0.02 | 0.49 ± 0.01 | **0.53 ± 0.03 I C** |
| | T8 | 0.69 ± 0.01 | 0.68 ± 0.02 | 0.69 ± 0.03 | 0.70 ± 0.04 |
| **MAS (m/min)** | T0 | 28 ± 9 | 32 ± 5 | 33 ± 6 | 31 ± 8 |
| | T8 | 23 ± 6 | **35 ± 11 S** | **33 ± 8 S** | **37 ± 3 S** |

Measurements are expressed as mean ± SD. **S**: p-value < 0.05 *vs* SED; **I**: p-value < 0.05 *vs* HIIT; **C**: p-value <0.05 *vs* CR. SED: Sedentary group; HIIT: High Intensity Interval Training group; CR: Continuous Running group; ComR: Combined Running group.

Intensity Interval Training group (HIIT) and the Combined Running group (ComR). The characteristics of each group are presented in Table 1

## 2.2. Experimental design

The rats followed an experimental treadmill running protocol approved by the Ethics and Animal Research Committee of Lariboisière/Villemin, Paris (Committee n°9) and by the French Ministry of Agriculture (APAFIS #9505). The age of the rats was seven weeks old at the beginning of the training protocol. At the beginning (T0) and end (T8) of the protocol, body weight and body length (nose to anus) were measured, and body mass index (BMI = body weight (g)/ length² (cm²)) calculated for each rat.

Before beginning the running protocols, the rats were subjected to a maximal aerobic speed (MAS) test on the fifth day of acclimatization, to determine their running speeds. The test began with a warm-up session at an incline of 10° and a speed of 13 m/min for 5 min, followed by increments of 4 m/min every 2 min until 17 min, then increments of 4 m/min every 1min30s. The rats were subjected to the test until they could no longer keep up with the speed of the treadmill, despite two successive stimulations with compressed air. The rat's maximum speed was defined as the last fully sustained incremental speed before cessation of the test. Once the exercise protocols had been completed, a MAS test was performed to validate the exercise programs.

The running protocols lasted eight consecutive weeks, after which the rats were euthanized. The three running groups underwent 45 minutes of training on a treadmill without incline every day, five days a week, for eight weeks. All groups underwent a five-minute warm-up at a speed of 12m/min before beginning their own protocol. The running speed chosen for the exercise protocol corresponds to the average MAS of each group.

For the rats in the CR group, the training session consisted of a 15 m/min run at 70% MAS. Rats in the HIIT group were trained in 7 blocks of 6 minutes each, alternating between medium intensity at 12m/min for 3 min and high intensity at 21 m/min (100% MAS) for 2 min followed by 1 min of rest. The ComR group followed the two running modalities during the training week. On Monday, Wednesday and Thursday, the rats ran continuously at 15m/ min for 40 minutes. On Tuesday and Friday, the rats ran in interval runs, in 7 blocks of 3 minutes at 12m/min followed by 2 minutes at 25m/min followed by 1-minute rest (Fig 1).

If necessary, the rats were encouraged to run by means of a compressed air jet pulse. The SED group had no activity other than moving around their cage.

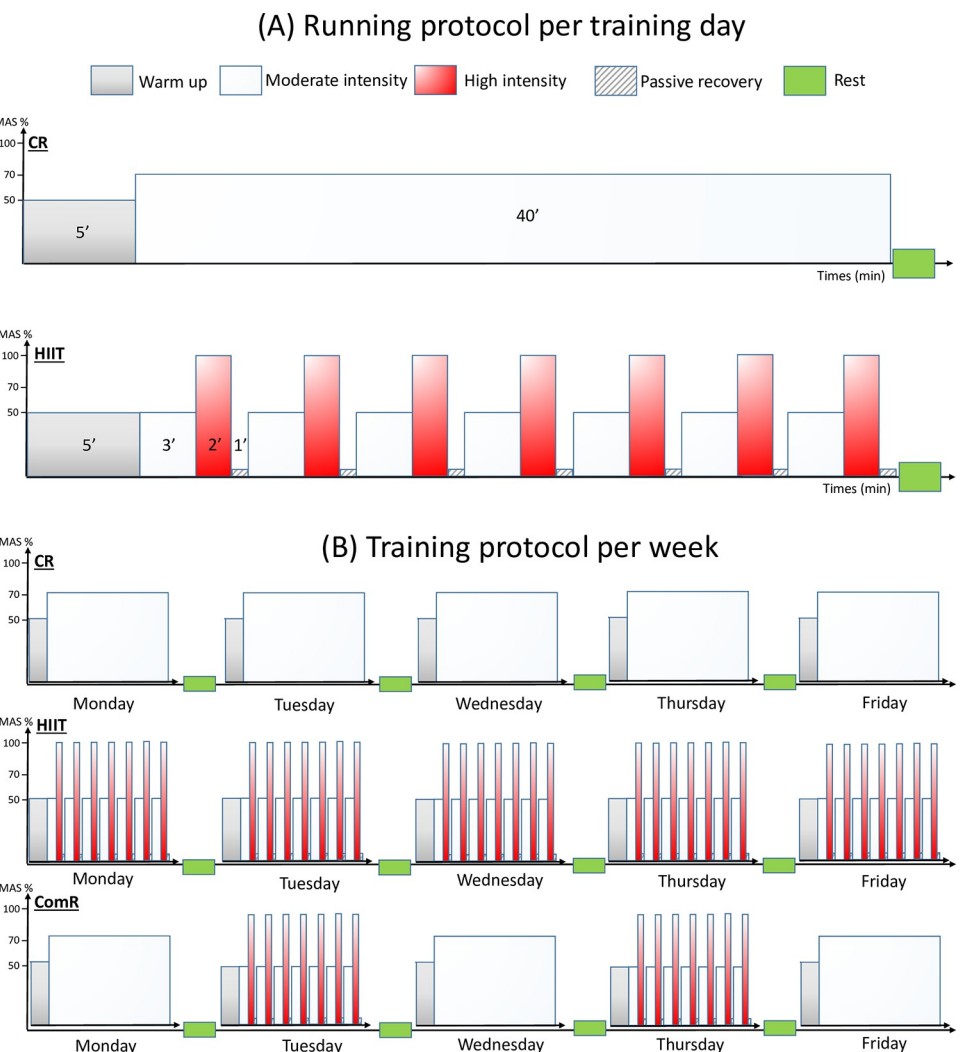

**Fig 1. Exercise protocols.** (A) During a day, rats, according to their group, either practiced continuous running (CR), or high intensity interval training (HIIT). (B) Organization of the 5 training days per week according to running protocols. CR: Continuous Running group; HIIT: High Intensity Interval Training group; ComR: Combined Running group (i.e. alternating between the continuous and the HIIT running).

To ensure that the level of intensity of running is equivalent and comparable, we have calculated training scores based on the work by Boudenot *et al.*, 2015 [19]; Mujika *et al.*, 1995 [17]. The equation is based on three coefficients according to speed (I#1: low speed $\leq$ 15 m/min, coefficient 1; I#2 moderate speed: 15 to 20 m/min, coefficient 2; I#3 intense speed: >20 m/min, coefficient 3). This score gives information about the difficulty of the exercise: the higher the score, the more difficult the exercise. The intensity score for each rat and then for each group was calculated as the average of the scores measured each week of running. The mean intensity score was 1.6±0.5 for the HIIT group and 1.7±0.7 for the CR group. No difference in score was observed between the two groups.

$$\text{Intensity score} = \frac{(\text{time spent at I\#1}) \times 1 + (\text{time spent at I\#2}) \times 2 + (\text{time spent at I\#3}) \times 3}{\text{total time}}$$

## 2.3. Tissue sampling

At the end of the 8-weeks running protocols, all animals were euthanized by exsanguination confirmed by cervical dislocation. The blood was collected in heparinized tubes, immediately cooled and centrifuged at 700g at 4˚C for 10 minutes, and the plasma was stored frozen at 80˚C until analysis. Radius, ulna and humerus were collected, cleaned of soft tissues and fixed in 4% v/v paraformaldehyde (PFA).

## 2.4. Bone morphological analysis

The morphology of the complete bones was evaluated using reconstructions of μCT sections at 9.81 μm resolution and analyzed on Dragonfly software (version 2022.2 Build 1399), in the coronal, axial and sagittal planes. The diameter of the medullary canal at the midshaft diaphysis of each bone was measured along the anteroposterior and mediolateral axis.

## 2.5. Cortical and trabecular bone microarchitecture measurements using *ex vivo* microcomputed tomography (μCT)

The left ulna, radius and humerus were scanned *ex vivo* using μCT (Bruker SkyScan 1176, Kontich, Belgium). The bones were positioned on a holder in a 1.8 mL cryotube rolled in polyethylene foam film with ethanol at 70˚. Two types of acquisitions were performed: a full scan at a 9.81 μm resolution and a mid-diaphysis scan at 4.91μm resolution to measure cortical porosity (measurement of the pores of the vascular channels). The X-rays source was set at a voltage of 80 kV with a current of 100 μA with an integration time of 175 ms for full bone and 375 ms for the cortical porosity. For all scans, a rotation step of 0.3˚, and a 0.5 mm aluminum filter were used. Acquired images were reconstructed using NRecon software (version 1.6.9.10, Bruker). The same parameters were applied to all the area from each bone site to ensure consistency (smoothing = 1, ring artefact = 20%, beam hardening = 100%). Regions of interest were selected and analyzed using Dragonfly software (version 2022.2 Build 1399) [24, 25]. Global thresholding was used to segment bone from the background. The upper threshold value was 255 and the lower threshold for each sample was assigned automatically using the Otsu threshold method [26]. The 9.81μm resolution acquisitions were analyzed in three modalities to assess the quality of the trabecular microarchitecture: full bone (whole bones were analyzed without the growth plates), proximal epiphysis (a region of interest consisting of 200 transversal sections from the distal end of the growth plate distal was selected), distal epiphysis (a region of interest consisting of 200 transversal sections from the proximal end of the growth plate proximal was selected). To examine the cortical bone at the midshaft diaphysis, a region of interest of 100 transversal sections was selected.

Following the initial pre-segmentation, the bone segmentation was filled corresponding to the bone cortex and marrow (filled = 0.2). The output of this process is a filled region of interest, which is an input to separate cortical and trabecular bone. Segmentation of cortical and trabecular bone was performed using the Buie method [27]. This method requires the thickness of the largest trabecular section of bone to be measured manually in order to supply the algorithm. For the data set, we set this parameter at 0.20 mm.

Following guidelines from Bouxsein *et al.*, 2010 [28], the following μCT parameters were reported. The trabecular outcome measures for the full bone, proximal epiphysis and distal epiphysis included the Bone Volume / Tissue Volume (BV/TV), Specific bone surface/ bone volume (BS/BV), Bone surface density (BS/TV), Trabecular Thickness (Tb.Th), Trabecular Number (Tb.N), Trabecular Spacing (Tb.Sp). Cortical outcome measures for the bone diaphysis included cortical bone area (Ct.Ar), total cross-sectional area (Tt.Ar), cortical area fraction

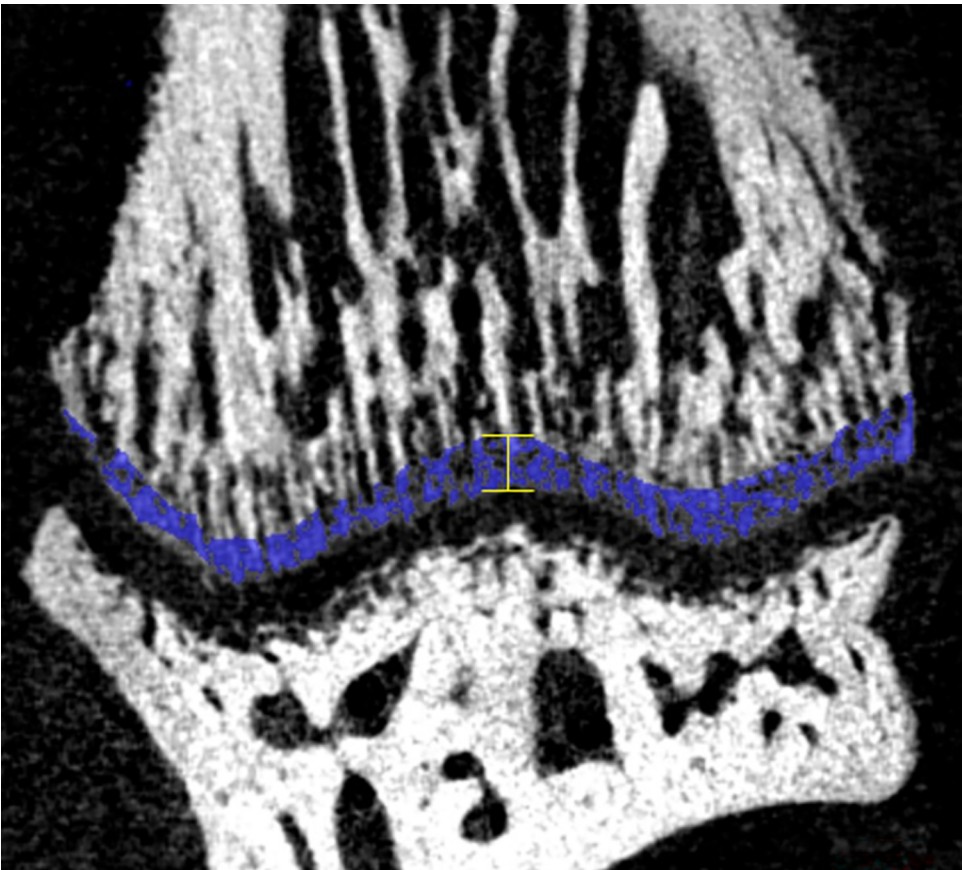

**Fig 2. Example of the subchondral bone selection on a cross-section of the distal ulna using Dragonfly software.** The area colored in blue corresponds to the subchondral bone selected according to the grey level. The yellow line corresponds to the measurement of the thickness of the subchondral bone.

calculated by cortical bone area over total area (Ct.Ar/Tt.Ar), and cortical thickness (Ct.Th). The dragonfly software has been used to calculate additional parameters, including the endo-cortical perimeter (Ec.Pm), the total surface area of the endocortex (Ec.S.3D), the mean surface area of the medulla (Ma.Ar), the total surface area of the periosteum (Ps.S.3D) and the mean total surface area of the cortical bone and marrow (Tt.Ar).

### 2.6. Subchondral bone thickness measurement

The thickness of the subchondral bone was measured in the middle of the distal condyles of radius and ulna at the *articulatio radiocarpalis* and at the *articulatio ulnocarpalis* using Drag-onfly software. Results are reported in millimeters (Fig 2).

### 2.7. Blood bone markers analysis

Total alkaline phosphatase (ALP), a marker of bone formation, was assayed using a standard laboratory analyzer (Selectra xl, Elitech) and an enzymatic kinetic kit (ALP (DEA) SL, Elitech), in the same run for all the samples, following the manufacturers' instructions. The limit of detection was equal to 6 U/L. Cross linked N-Telopeptide of type I collagen (NTx), a marker of bone resorption, was measured using established rat enzyme linked immunosorbent assay kits (Antibodies-online GmbH, Aachen, Germany [29]), following the manufacturers'

instructions. The intra-/interassay coefficients of variation and limit of detection for NTx were <12% and 2.47 ng/mL respectively.

## 2.8. Statistical analysis

Statistical analyses were performed using a commercially available software (GraphPad Prisme, Version 8.0.2). Descriptive statistics were reported as means ± standard deviation (SD). The normality of the data distribution was performed with the Shapiro–Wilk test and the homogeneity of variances was performed with the Leven test. Because normality of the data distribution was not achieved, therefore non parametric tests were used: first Kruskall Wallis test to compare all 4 groups then U Mann–Whitney test for comparison between two specific groups. The level of significance was set at p-value < 0.05.

# Results

## 3.1. Body parameters, food and water consumption

Body parameter results are summarized in Table 1. At the end of the protocol, the body mass index, weight, and body length were not statistically different between SED and the three running groups. Food and water consumptions were not different between SED and all running groups (mean food: 25.1 ± 4.1 and 27.6 ± 2.9 g, for SED and running groups rats respectively; mean water: 31.4 ± 3.1 and 37.6 ± 11.3 mL for SED and all running groups respectively).

## 3.2. MAS

At the beginning of the running protocol, no statistically significant differences in MAS were observed among groups. After 8 weeks of training (T8), an increase in MAS was observed in all running groups, whereas the MAS in the SED group remained similar to the beginning of the study (T0) (Table 1), these results validate the training protocol.

## 3.3. Morphological examination

Morphological measurements of the radius, ulna and humerus in each group are presented in Table 2. No differences in bone length (proximal-distal) were found. The results have shown that 8 weeks of training did not interfere with bone growth, whatever the type of running modality.

In the radius, ComR resulted in a significant increase in the mediolateral diameter (7.5% and 5.4% compared to sedentariness and CR, respectively). It also resulted in a significant 29% increase in the antero-posterior diameter of the medullary canal compared to SED. In the ulna, HIIT resulted in a significant increase in mediolateral diameter (9%) compared to SED. CR and ComR resulted in a significant increase in the mediolateral diameter of the medullary canal (37% and 45%, respectively), compared to SED. In the humerus, a significant decrease (13%) in the mediolateral diameter of the medullary canal in the HIIT group was reported, compared to the CR group. In contrast, ComR resulted in an increase in the mediolateral diameter of the medullary canal, compared to both SED and HIIT (8% and 13%, respectively).

## 3.4. Bone trabecular and cortical microarchitecture measurements by µCT in radius, ulna and humerus

**3.4.1. Radius.** The µCT measurements in the radius are presented in supplementary files (S1 Table). No significant differences were found in trabecular microarchitectural parameters (BV/TV, BS/BV, Tb.N, Tb.Sp, Tb.Th) among groups. Differences in bone and medullary morphology, however, were observed (Table 3). An increase in the mean cortical surface fraction

**Table 2. Morphological measurements of the radius, ulna and humerus limbs according to the running protocols.**

| Bones | Morphological measurements (mm) | SED | HIIT | CR | ComR |
|---|---|---|---|---|---|
| **Radius** | Proximal-distal length | 25.49 ± 0.60 | 25.38 ± 0.48 | 25.54 ± 0.53 | 25.40 ± 0.65 |
| | Antero-posterior diameter | 1.74 ± 0.10 | 1.68 ± 0.11 | 1.67 ± 0.09 | 1.78 ± 0.18 |
| | Medio-lateral diameter | 1.99 ± 0.10 | 2.10 ± 0.25 | 2.03 ± 0.09 | **2.14 ± 0.10 S C** |
| | Anterior cortical thickness | 0.60 ± 0.04 | 0.63 ± 0.07 | 0.62 ±0.08 | 0.62 ± 0.07 |
| | Posterior cortical thickness | 0.50 ± 0.07 | 0.50 ± 0.09 | 0.56 ±0.06 | 0.54 ± 0.08 |
| | Medial cortical thickness | 0.76 ± 0.09 | 0.70 ±0.09 | 0.64 ± 0.09 | 0.65 ± 0.12 |
| | Lateral cortical thickness | 0.44 ± 0.04 | 0.42 ± 0.04 | 0.45 ± 0.04 | 0.46 ± 0.04 |
| | MC medio-lateral diameter | 0.36 ± 0.07 | 0.37 ± 0.10 | 0.40 ± 0.08 | 0.43 ± 0.07 |
| | MC antero-posterior diameter | 0.58 ± 0.05 | 0.61 ± 0.12 | 0.66 ± 0.10 | **0.75 ± 0.12 S** |
| **Ulna** | Proximal-distal length | 33.00 ± 0.58 | 32.92 ± 0.59 | 33.10 ± 0.71 | 33.10 ± 0.60 |
| | Antero-posterior diameter | 2.74 ± 0.13 | 2.75 ± 0.14 | 2.82 ± 0.02 | 2.75 ± 0.06 |
| | Medio-lateral diameter | 1.14 ± 0.09 | **1.24 ± 0.06 S** | **1.23 ± 0.07 S** | 1.21 ± 0.04 |
| | Anterior cortical thickness | 1.19 ± 0.16 | 1.22 ± 0.07 | 1.15 ± 0.08 | 1.18 ± 0.07 |
| | Posterior cortical thickness | 0.96 ± 0.06 | 0.91 ± 0.07 | 0.89 ± 0.13 | 0.86 ± 0.08 |
| | Medial cortical thickness | 0.48 ± 0.03 | 0.49 ± 0.02 | 0.48 ± 0.02 | 0.48 ±0.02 |
| | Lateral cortical thickness | 0.41 ±0.03 | 0.38 ±0.03 | 0.39 ±0.03 | 0.41 ± 0.05 |
| | MC medio-lateral diameter | 0.165 ± 0.044 | 0.195 ± 0.043 | **0.226 ± 0.049 S** | **0.239 ± 0.028 S I** |
| | MC antero-posterior diameter | 0.721 ± 0.115 | 0.707 ± 0.103 | 0.767 ± 0.069 | 0.780 ± 0.077 |
| **Humerus** | Proximal-distal length | 29.62 ± 0.53 | 29.36 ± 0.49 | 29.70 ± 0.67 | 29.84 ± 0.59 |
| | Antero-posterior diameter | 4.54 ± 0.57 | 4.34 ± 0.56 | **4.97 ± 0.36 I** | 4.63 ± 0.36 |
| | Medio-lateral diameter | 2.68 ± 0.14 | 2.70 ± 0.13 | 2.71 ± 0.15 | 2.78 ± 0.07 |
| | Anterior cortical thickness | 0.83 ± 0.11 | 0.83 ± 0.10 | 0.83 ± 0.04 | 0.84 ± 0.10 |
| | Posterior cortical thickness | 0.57 ± 0.07 | 0.57 ± 0.06 | 0.61 ±0.06 | 0.56 ± 0.06 |
| | Medial cortical thickness | 0.55 ± 0.03 | 0.54 ± 0.05 | 0.55 ± 0.03 | 0.55 ± 0.04 |
| | Lateral cortical thickness | 0.57 ± 0.05 | 0.57 ± 0.05 | 0.55 ± 0.04 | 0.54 ± 0.04 |
| | MC medio-lateral diameter | 1.57 ± 0.33 | 1.50 ± 0.08 | 1.60 ± 0.12 | **1.70 ± 0.16 S I** |
| | MC antero-posterior diameter | 1.98 ± 0.22 | 1.97 ± 0.16 | 1.86 ± 0.24 | 2.11 ± 0.15 |

Measurements are expressed as mean ± SD in mm. Proximal-distal length was measured between the proximal and distal epiphyses of each bone, and bone diameters (antero-posterior, medio-lateral, MC medio-lateral, MC antero-posterior) were measured at the mid-diaphysis. MC = medullary canal. **S**: p-value <0.05 *vs* SED; **C**: p-value <0.05 *vs* CR; **I**: p-value < 0.05 *vs* HIIT. SED: Sedentary group; HIIT: High Intensity Interval Training group; CR: Continuous Running group; ComR: Combined Running group.

was reported in the HIIT group, compared to respective results in the ComR (5% increase) and CR (4% increase) groups. Similar results were obtained for the endocortical perimeter (Ec. Pm), the total endocortex surface area (Ec.S.3D) and the mean medullary surface area (Ma. Ar). Compared to ComR, HIIT also induced a 5% decrease in the total periosteal surface area (Ps.S.3D) and a 8% decrease in the mean total surface area of cortical bone and marrow (Tt. Ar).

**3.4.2. Ulna.** In the ulna, differences in trabecular microarchitecture were only observed in the distal metaphysis (S2 Table). More specifically, compared to the SED groups, decreases in BV/TV of 11% and 8% were reported in the HIIT and CR groups, respectively. Tb.Th in the ComR group was increased by 11% and 24% compared to respective results in the HIIT and CR groups.

The mean cortical surface fraction (Ct.Ar/Tt.Ar) in the HIIT group was increased, compared respective results in all the other groups (Table 3). This result may be explained by the decrease in endocortical perimeter (EC.Pm), endocortical surface area (Ec.S.3D), and mean

**Table 3. Cortical bone microarchitectural parameters and medullary canal morphology analysis by µCT in the radius, ulna, and humerus according to the running protocols.**

| Cortical bone microarchitectural parameters | | SED | HIIT | CR | ComR |
|---|---|---|---|---|---|
| Radius | Ct.Ar/Tt.Ar (%) | 0.731 ± 0.024 | 0.743 ± 0.034 | **0.714 ± 0.021 I** | **0.710 ± 0.027 I** |
| | Ec.Pm (mm) | 3.40 ± 0.26 | 3.22 ± 0.33 | **3.58 ± 0.24 I** | **3.68 ± 0.25 I** |
| | Ec.S.3D (mm$^2$) | 101 ± 8 | 94 ± 10 | **106 ± 8 I** | 109 ± 7 |
| | Ma.Ar (mm$^2$) | 0.67 ± 0.07 | 0.62 ± 0.11 | **0.72 ± 0.08 I** | **0.76 ± 0.09 S I** |
| | Ps.S.3D (mm$^2$) | 166.8 ± 4.8 | 161.5 ± 7.4 | 167.1 ± 5.7 | **169.4 ± 4.0 I** |
| | Tt.Ar (mm$^2$) | 2.49 ± 0.15 | 2.38 ± 0.16 | **2.52 ± 0.12 I** | **2.60 ± 0.08 I** |
| Ulna | Ct.Ar/Tt.Ar (%) | 0.689 ± 0.013 | **0.720 ± 0.018 S** | **0.682 ± 0.026 I** | **0.695 ± 0.021 I** |
| | Ec.Pm (mm) | 4.86 ± 0.22 | **4.41 ± 0.29 S** | **4.98 ± 0.25 I** | **4.80 ± 0.30 I** |
| | Ec.S.3D (mm$^2$) | 175 ± 11 | **158 ± 12 S** | **182 ± 9 I** | **174 ± 12 I** |
| | Ma.Ar (mm$^2$) | 0.99 ± 0.07 | **0.86 ± 0.09 S** | **1.05 ± 0.10 I** | **0.98 ± 0.10 I** |
| | Ps.S.3D (mm$^2$) | 249.5 ± 8.2 | 243.5 ± 10.9 | 251.3 ± 7.2 | 253.5 ± 6.8 |
| | Tt.Ar (mm$^2$) | 3.18 ± 0.14 | 3.07 ± 0.18 | 3.31 ± 0.09 | **3.22 ± 0.13 S I** |
| Humerus | Ct.Ar/Tt.Ar (%) | 0.481 ± 0.023 | 0.477 ± 0.022 | 0.475 ± 0.022 | 0.458 ± 0.024 |
| | Ec.Pm (mm) | 10.64 ± 0.45 | 10.36 ± 0.39 | 10.70 ± 0.51 | **10.91 ± 0.22 I** |
| | Ec.S.3D (mm$^2$) | 349 ± 18 | 336 ± 17 | 349 ± 19 | **359 ± 11 I** |
| | Ma.Ar (mm$^2$) | 4.58 ± 0.35 | 4.45 ± 0.34 | 4.68 ± 0.36 | **4.97 ± 0.30 S I** |
| | Ps.S.3D (mm$^2$) | 396.7 ± 15.5 | 388.2 ± 11.6 | 398.9 ± 13.4 | 406.5 ± 10.5 |
| | Tt.Ar (mm$^2$) | 8.82 ± 0.36 | 8.49 ± 0.39 | **8.90 ± 0.38 I** | **9.17 ± 0.26 S I** |

Values are expressed as mean ± SD, obtained from µCT acquisition (Bruker SkyScan 1176, Kontich, Belgium) and analyzed with DragonFly software (version 2022.2 Build 1399). **S**: p-value <0.05 *vs* SED; **I**: p-value < 0.05 *vs* HIIT. SED: Sedentary group; HIIT: High Intensity Interval Training group; CR: Continuous Running group; ComR: Combined Running group. Ct.Ar/Tt.Ar: Cortical Area/ Total Area is the ratio of the mean cortical surface to the mean total surface; Ec.Pm: Endocortical Perimeter; Ec.S.3D: Endocortical Surface 3D is the measure of total endocortex surface area; Ma.Ar: Marrow Area is the mean medullary surface area; Ps.S.3D: total surface area of the periosteum; Tt.Ar: Total Area is the mean total surface area of cortical bone and marrow.

medullary surface area (Ma.Ar). These results confirm the morphologic evaluation, in which the diameter of the medullary canal in the ulna obtained from rats subjected to HIIT was decreased compared to the respective results for ulna obtained from rats subjected to ComR (Table 3, Fig 3).

**3.4.3. Humerus.** No difference in BV/TV, Tb.Th, Tb.Sp parameters was observed in humerus obtained from all the experimental groups (S3 Table). Similarly to what was observed in the ulna, the endocortical perimeter (EC.Pm), endocortical surface area (Ec.S.3D) and mean medullary surface area (Ma.Ar) in the HIIT group were decreased, compared to respective results in the ComR group (Table 3). Humerus obtained from rats subjected to HIIT had a significantly lower periosteal perimeter than humerus obtained from rats subjected the CR and ComR (by 2% and 3%, respectively).

## 3.5. Cortical porosity in radius, ulna and humerus

The cortical porosity parameters for the radius, ulna and humerus obtained from each group are presented in Table 4. In the radius obtained from the CR group, the pore number (Po.N) was 75% upper than results from the SED group, 74% and 87% upper than respective results from both HIIT and ComR group. The pore diameter (Po.Dm) was 40% lower than results observed in SED group. In the ulna obtained from the HIIT group, the cortical porosity (Ct.

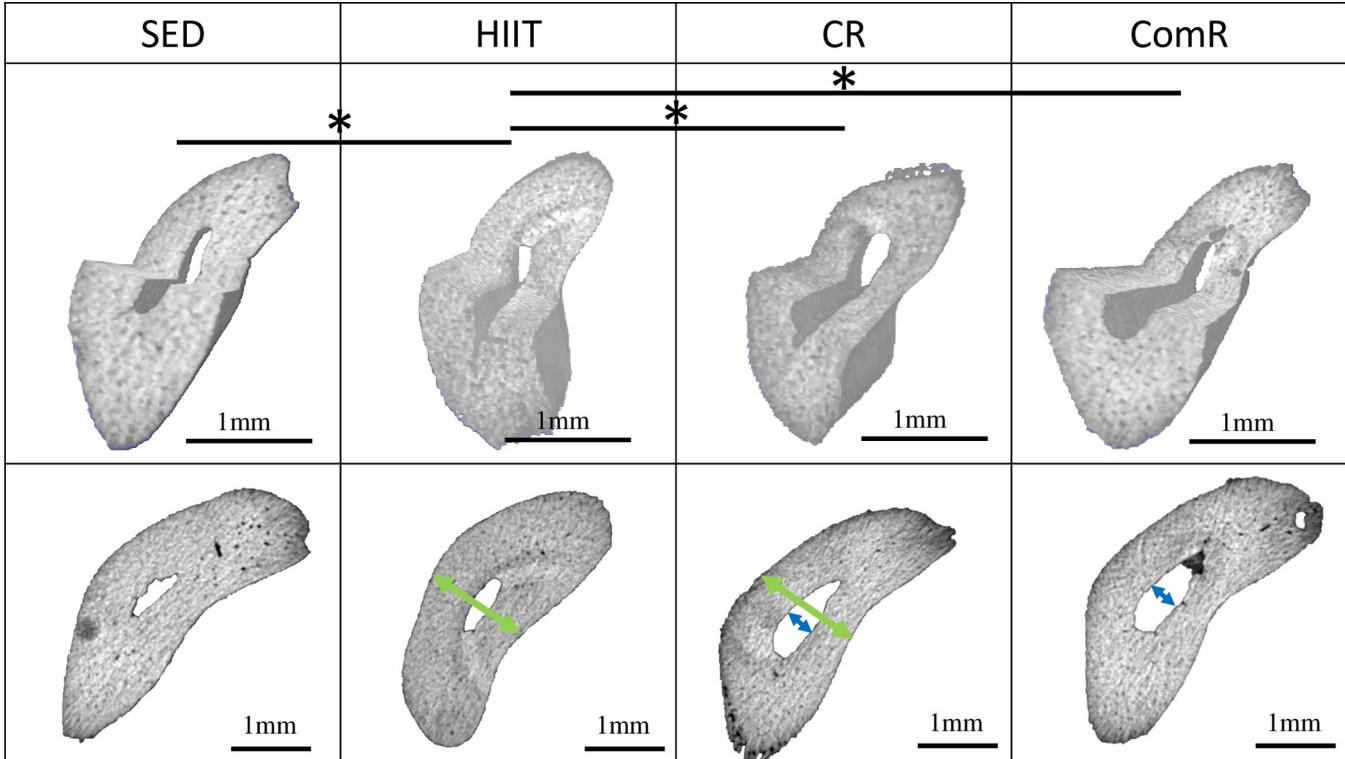

**Fig 3. Illustration of the main differences in ulna morphology between running groups, based on morphometric and microarchitecture analyses.** Images acquired with Bruker SkyScan 1176 (Kontich, Belgium) and analyzed by DragonFly software (version 2022.2 Build 1399). The upper images represent the three-dimensional reconstruction of the ulna at mid-diaphysis. The upper line of table corresponds to the differences in the analysis of the cortical microarchitecture (Ct.Ar/Tt.Ar: Cortical Area/Total Area) of the ulna at the mid-diaphysis. The lower images represent the ulnar cortices at mid-diaphysis. The lower line of table corresponds to the morphometric measurements of the ulnar cortical at mid-diaphysis (the arrows correspond to the significant difference compared to the SED group: green arrow: medio-lateral diameter, blue arrow: medio-lateral diameter of the medullary canal). SED: Sedentary group; HIIT: High Intensity Interval Training group; CR: Continuous Running group; ComR: Combined Running group. The statistically significant differences (*p-value<0.05) are summarized in Tables 2 and 3.

**Table 4. Cortical porosity at mid diaphysis analyzed by µCT of the radius, ulna and humerus with respect to running modality.**

| Cortical porosity parameters | | SED | HIIT | CR | ComR |
|---|---|---|---|---|---|
| **Radius** | Ct.Po (%) | 2.17 ± 0.22 | 2.27 ± 0.37 | 2.15 ± 0.27 | 2.01 ± 0.12 |
| | Po.N (mm$^{-1}$) | 4.99 ±0.27 | 5.02 ± 0.44 | **8.71 ± 2.64 S I B** | 4.66 ± 0.44 |
| | Po.Sp (mm) | 0.01 ± 0.00 | 0.01 ± 0.00 | 0.01 ± 0.00 | 0.01 ± 0.00 |
| | Po.Dm (mm) | 0.20 ± 0.01 | 0.20 ± 0.02 | **0.12 ± 0.04 S I B** | 0.21 ± 0.02 |
| **Ulna** | Ct.Po (%) | 2.24 ± 0.16 | **3.70 ± 0.96 S B** | 3.50 ± 1.49 | 2.11 ±0.16 |
| | Po.N (mm$^{-1}$) | 5.22 ± 0.40 | **13.43 ± 3.78 S B** | 10.85 ± 5.45 | 5.33 ± 0.24 |
| | Po.Sp (mm) | 0.01 ± 0.00 | 0.01 ± 0.00 | 0.01 ± 0.00 | 0.01 ± 0.00 |
| | Po.Dm (mm) | 0.19 ± 0.01 | **0.08 ± 0.04 S B** | 0.12 ± 0.06 | 0.18 ± 0.01 |
| **Humerus** | Ct.Po (%) | 2.32 ± 0.44 | 2.32 ± 0.32 | 2.10 ± 0.34 | 2.27 ± 0.52 |
| | Po.N (mm$^{-1}$) | 11.75 ±0.96 | **9.56 ± 1.43 S** | 10.91 ± 1.09 | 11.00 ± 1.20 |
| | Po.Sp (mm) | 0.01 ± 0.00 | 0.01 ± 0.00 | 0.01 ± 0.00 | 0.01 ± 0.00 |
| | Po.Dm (mm) | 0.09 ± 0.02 | 0.10 ± 0.01 | 0.09 ± 0.01 | 0.09 ± 0.01 |

Values are expressed as mean ± SD, obtained from µCT acquisition (Bruker SkyScan 1176, Kontich, Belgium) and analysed with DragonFly software (version 2022.2 Build 1399). **S**: p-value <0.05 *vs* SED; **C**: p-value < 0.05 *vs* CR. **B**: p-value < 0.05 *vs* ComR. SED: Sedentary group; HIIT: High Intensity Interval Training group; CR: Continuous Running group; ComR: Combined Running group. Ct.Po: Cortical Porosity; Po.N: Pore Number; Po.Sp: Pore Spacing; Po.Dm: Pore Diameter.

Po) was 65% and 75% greater than respective results from the SED and ComR group. The pore number (Po.N) was 157% and 152% upper than respective results from the SED and ComR group. The pore diameter (Po.Dm) was 58% and 56% lower than respective results from the SED and ComR group. For the humerus, the HIIT group had a lower (19%) pore number (Po.N) than that observed in the SED groups.

### 3.6. Subchondral bone thickness

An increase in the radial subchondral bone thickness at the *articulatio radiocarpalis* by 32% and 51% in the HIIT and CR groups respectively, was measured compared to the SED group. An increase in ulnar subchondral bone thickness was measured in all running groups (by 95%, 69%; and 19% in the HIIT, CR and ComR groups, respectively) compared to the SED group (Fig 4).

### 3.7. Bone turnover markers

Plasma analyses of bone remodeling markers (ALP, NTX) at the end of the protocol showed no significant difference among the groups.

## Discussion

The aim of the present study was to compare the effects of three different treadmill running modalities on forelimb bones in a male Wistar rat model (Fig 5). The studies carried out in the forelimbs in the literature have proposed models of resistance exercise in terms of force (pulling on a force lever) [30, 31], constraint (forearm compression) [32] or continuous running [33]. In quadrupeds, the function of the forelimbs is to cushion the impact when running, whereas the function of the hindlimbs is to propel the body, which resulted in an increase in stress on the ulna and radius. The study by Wallace *et al.*, 2015, [23] compared the effects of continuous running in a mouse model in all four limbs. Their biomechanical study on a force platform showed that the forelimbs were subjected to higher-intensity impacts than the hindlimbs. However, they showed through trabecular microarchitecture analyses that the bones in mouse forelimb bones were less sensitive to exercise than the bones in the hind limbs [23]. In rodents, it appears that there are no changes in the bone quality in forelimb, in trabecular microarchitecture specifically, in response to gripping and object handling (food) tasks. This hypothesis is supported by studies in birds. Although the forelimbs are subjected to enormous stress during flight, the trabecular microarchitecture of their skeleton has adapted to maintain a light structure and has a lower volume fraction than that of land animals [34].

The 8-weeks protocol led to an increase in MAS in all running groups compared with the SED group. This increase is in line with our expectations, since it has been shown in the literature that endurance training improves MAS and maximum oxygen consumption [35]. The increase in MAS in our running groups validates our treadmill running protocols.

Morphological measurements of the bones in the forelimbs obtained from each experimental group showed that 8 weeks of training did not interfere with bone growth, whatever the type of race, since no difference in bone length was measured. These results are consistent with the weight recordings, in which no difference was observed. Morphometric measurements revealed an increase in the mediolateral diameter of the ulna from rats subjected to HIIT and CR., Radius obtained from rats in the ComR group presented an increase in their mediolateral diameter. The internal diameter of the medullary canal in the mediolateral axis was also increased in both the ulna and humerus in the ComR group compared to the SED and HIIT groups. This increase in diameter was also found in the radius in its antero-posterior axis. The measurements were performed using the axis of the rat's body as the theoretical

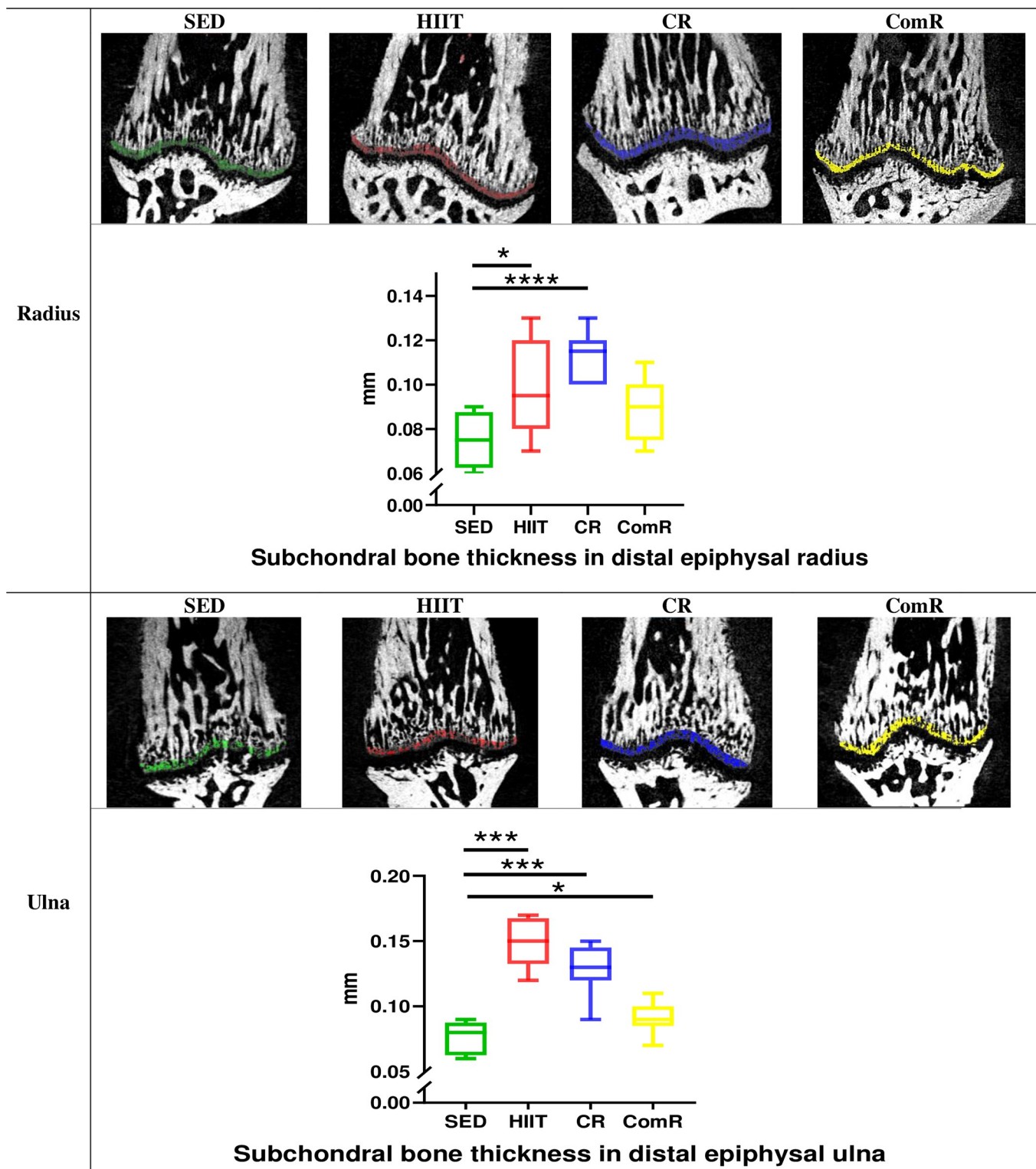

**Fig 4. Subchondral bone thickness in the distal radius and ulna.** Measurements were performed with Dragonfly software from μCT images. Values are expressed in mm. *: p-value < 0.05; ***: p-value < 0.001. SED: Sedentary group (green); HIIT: High Intensity Interval Training group (red); CR: Continuous Running group (blue); ComR: Combined Running group (yellow).

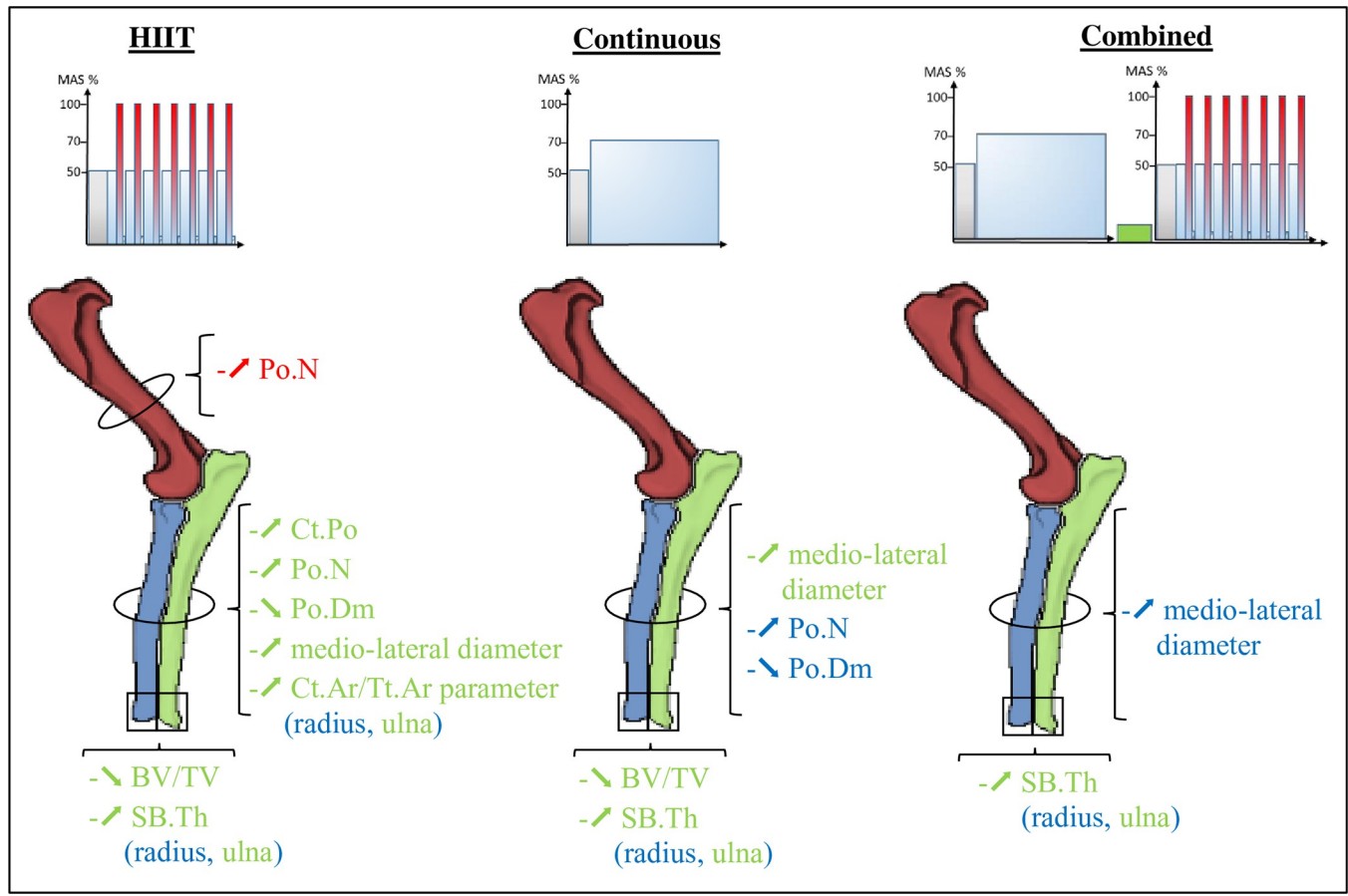

**Fig 5. Summary schematic of the main statistically significant results of the effects of the three running modalities on the upper limbs compared with the sedentary group.** (humerus in red, radius in blue, ulna in green); HIIT: High Intensity Interval Training group. ↘: Decrease; ↗: Increase; Ct.Po: Cortical Porosity; Po.N: Pore Number; Po.Dm: Pore diameter; BV/TV: Bone Volume/Tissue Volume; SB.Th: Subchondral Bone Thickness.

reference frame, according to anatomical representations in the literature. However, when the rat is walking, the radius undergoes a translation and a medial rotation so that it is completely crossed over the ulna when it rests on the ground. This rotation was measured at around 174–178° compared to a theoretical reference position during an analysis of the rat's walking [36]. This means that at the time of weight-bearing, the initial axis of the antero-posterior radius would potentially be in the mediolateral axis. The increase in the bone diameter at mid-diaphyseal level is probably a tissue response to the mechanical stresses associated with exercise, increasing the resistance of bone to flexion and torsion [37]. This phenomenon has been evaluated in athletes [38, 39] particularly in tennis players during and after puberty [40], as well as in continuous running models in rats [16] and in the forearm axial compression loading model [41].

The microarchitecture of the forelimb bones was analyzed in three regions (proximal epiphysis, distal epiphysis and whole bone) because the study aimed to determine whether the effects of impact resulted in local adaptation (epiphysis) or an adaptation of the whole bone. From a biomechanical point of view, the ulna appeared to be the bone that received the most stress, as it transmits the motor force through the insertion of the Musculus Triceps brachii. Analysis of the trabecular microarchitecture parameters [28] (including BV/TV, Tb.Th) revealed a difference between running groups, in the distal ulna only. Specifically, ComR had

the best osteogenic effects on trabecular bone microarchitecture at the distal ulnar epiphysis, with the highest value of the Tb.Th. parameter. The femur study by Wazzani *et al.*, 2023 [22] revealed that combined running improved trabecular microarchitecture parameters (Tb.Th; Tb.BMD) in the proximal femur. Here, specific analysis of the proximal and distal epiphyses of the radius and humerus revealed no significant difference. In the study herein, ComR running increased the internal medullary cavity area (Ma.Ar) in radius and humerus, compared to respective results in the SED and HIIT groups, in ulna compared to respective results in the HIIT condition. These results are consistent with morphometric measurements of medullary canal diameter at mid-diaphysis. In the ulna and radius, these results are reflected in a decrease in the average cortical surface fraction (Ct.Ar/Tt.Ar) in the ComR group, compared to the HIIT group. It therefore seems that combined running increases the diameter of the radius and ulna, while reducing the cortical surface. This adaptation is described in the literature during bone growth, when endocortical resorption results in an expanding medullary cavity and partly compensates for the increase in cortical surface due to periosteal apposition [37]. The consecutive result is an increase in the diameter of the cortical bone, which results in an increase in its resistance to flexion [37]. In the ulna, combined running therefore appears to have positive effects on distal trabecular microarchitecture (BV/TV, Tb.Th) as well as on the theoretical flexural strength of the diaphysis. These results are consistent with previous studies, in which young growing rats (Sprague-Dawley rats, 2.5 months of age) showed an improvement in trabecular microarchitecture parameters in the distal radius (BV/TV; Tb.Th; Tb.N) and an increase in medullary cavity area, associated with improved bone tensile strength [42] https://www.ncbi.nlm.nih.gov/pmc/articles/PMC4655973/. However, in the same study, the strength protocol also led to a degradation of the trabecular and cortical bone in adult rats. The authors concluded that excessive and repeated mechanical loadings were beneficial in young rats but presented deleterious effects in adult rats.

Assessment of cortical porosity (Ct.Po) by μCT at 4.81μm resolution (vascular channels) reflects the mechanical strength of the bone [43]. This network of channels evolves as the bone develops to promote vascularization and blood perfusion and ensure the adequate supply of oxygen, minerals and nutrients required for bone homeostasis [44] and growth [45]. Our study revealed significant differences in cortical porosity at the ulna, the bone that absorbs mechanical stress and transmits muscular force during locomotion. At the ulna, the HIIT group had greater cortical porosity than the sedentary and combined running groups. Although the literature on the effects of exercise on bone health is widely developed, the actual impact of interval running on bone tissue is not yet fully understood [46]. In this respect, it has been observed in humans that high-intensity training can generate a significant loss of cortical bone, favoring the appearance of injuries or fractures [47, 48], but it has also been shown that in untrained subjects high-intensity interval running increases bone density measured in the whole body by DXA [49]. Studies in humans seem to lack homogeneity in terms of the parameters evaluated and the results reported (blood dosage and bone density), which means that no conclusions can be drawn. Nevertheless, in male mouse models, the response of bone tissue to high intensity running, either performed continuously or at intervals, appears to increase the fraction of cortical bone to the detriment of mechanical strength [48, 50]. These two studies also showed that the response of bone tissue is also dependent on the sites where measurements were performed and sexual dimorphism, with the tibias of male mice showing the greatest effects [48, 50]. Thus, speed or intensity would condition the effects on bone tissue, and it has been shown in rats (5-month-old male Sprague Dawley) that at 12m/min (continuous) running did not induce any variation in cortical porosity, whereas at 16m/min running induced an increase in cortical porosity [51]. In the radius, CR decreased the number of cortical pores (Po.N) compared with the sedentariness. CR increased pore diameter (Po.Dm)

compared with the ComR. To date, no study has compared the effects of these three running protocols on forelimb cortical porosity. However, studies that have applied compressive stress [52] or force (traction) protocols [42] have shown deleterious effects on the rate of osteocyte lacunar occupation at the mid-diaphysis, increased by cortical micro-fractures and an increase in cortical porosity in the radius. The studies which carried out their analysis on the tibias showed contradictory results. It was shown that continuous running had no effect on morphometry at the cortical diaphysis in sprague dawley rats (17m/min for 1 H 5 day/week for 9 weeks) (cortical bone area, marrow cavity area) [53] and in C57BL/6J mice, (12m/min at 5˚ incline for 30 min, 5 day/week for 6 weeks) (cortical thickness) [54] 10.1016/8756-3282(94) 90294-1 whereas other studies have shown that continuous running decreased cross sectional area with deleterious effects on mechanical properties (ultimate deformation) (C57BL/6J mice, 12 m/min, a 5˚ incline, 30 min/day for 8 weeks) [55] and on microarchitecture parameters (Po.Sp) and mediolateral tibial external diameter (Wistar rat, 20m/min for 60mins, 5 days a week for 14 weeks) [16].

The secondary aim of the study was to determine the effects of this three running protocols on the carpal and elbow joints, and more specifically the effects on subchondral bone. All three running protocols led to a thickening of the subchondral bone at the distal epiphysis of the radius and ulna. According to these results, interval or continuous trainings could have deleterious effects on the *articulatio radiocarpalis* and the *articulatio ulnocarpalis*. This is in line with the literature on osteoarthritis, in which interval running at high intensity induced thickening of the subchondral bone at the level of the tibial plateau in a rat model [56]. The study by Li *et al.*, 2017 [57] shows that continuous high-intensity running induces a change in the quality of cartilage and leads to a change in the organization of the microstructure and composition of subchondral bone in healthy Wistar rats. This study also shows that high-intensity physical activity alters both the tissue and the mechanical properties of bone and cartilage tissues concomitantly [57]. This concomitant alteration is the result of bone adaptation to mechanical stress, which leads to an increase in tissue density and hardness, which in turn leads to an increase in shear forces on the cartilage. Cartilage altered in this way is no longer able to maintain its viscoelastic properties and its role as a damper, leading to an increase in stress on the subchondral bone as it tries to adapt.

The three running protocols proposed in our study did not significantly modify the systemic markers of bone remodeling (ALP and NTx). To our knowledge, only one study has proposed similar protocols with three running modalities and did not reveal any differences in bone remodeling markers among groups [22]. In 6-month-old ovariectomized rat models, however, continuous moderate running (13 m/min for 60 min/day 5 days/week for 9 weeks) induced a decrease in CTX (C-terminal telopeptide) compared with the sedentary ovariectomized groups [14], whereas in 8-week-old male Wistar rats, continuous moderate running (22.5 m/min for 40–45 min 5 days/week for 8 weeks) induced an increase in CTX compared with the sedentary control group [15]. Studies in male Wistar rats showed that HIIT reduced NTX levels compared with the sedentary group [58].

## Conclusion

This study shows that continuous running, interval running and combined running led to bone adaptation resulting in an increase in the diameters of the bone shafts, and an adaptation of the microarchitecture in the ulna The combined running protocol, which aimed to mimic as closely as possible the training of high-level athletes, appears to have positive effects on the trabecular microarchitecture in the distal ulna (specifically Tb.Th. and Ma.Ar parameters). Interestingly, all three running modalities led to a thickening of the subchondral bone in the

distal epiphysis for both radius and ulna. The three running protocols did not induce any variation in plasma bone biomarker levels. Based on the results, HIIT is the running modality that induces the most bone adaptation with an increase in cortical area at the ulna and tibia associated with an increase in cortical porosity. Furthermore, regarding our results, it would appear that the bone response to the mechanical stress of running is essentially located on the distal epiphyses of the radius and ulna, i.e. close to the point of impact, which supports Wolff's law.

## Supporting information

**S1 Table. Trabecular microarchitecture analysis by μCT of the radius as a function of running modality.** Measurements are expressed as mean ± SD, measured by μCT (Bruker SkyScan 1176, Kontich, Belgium) and analyzed by DragonFly software (version 2022.2 Build 1399). SED: Sedentary group; HIIT: High Intensity Interval Training group; CR: Continuous Running group; ComR: Combined Running group. BV/TV: Bone Volume/Tissue Volume; Tb.N: Trabecular Number; Tb.Sp: Trabecular Spacing; Tb.Th: Trabecular Thickness.
(PDF)

**S2 Table. Trabecular microarchitecture analysis by μCT of the ulna as a function of running modality.** Measurements are expressed as mean ± SD, measured by μCT (Bruker SkyScan 1176, Kontich, Belgium) and analyzed by DragonFly software (version 2022.2 Build 1399). S: p-value <0.05 vs SED; C: p-value <0.05 vs CR; I: p-value < 0.05 vs HIIT. SED: Sedentary group; HIIT: High Intensity Interval Training group; CR: Continuous Running group; ComR: Combined Running group. BV/TV: Bone Volume/Tissue Volume; Tb.N: Trabecular Number; Tb.Sp: Trabecular Spacing; Tb.Th: Trabecular Thickness.
(PDF)

**S3 Table. Trabecular microarchitectural analysis by μCT of the humerus as a function of running modality.** Measurements are expressed as mean ± SD, measured by μCT (Bruker SkyScan 1176, Kontich, Belgium) and analyzed by DragonFly software (version 2022.2 Build 1399). SED: Sedentary group; HIIT: High Intensity Interval Training group; CR: Continuous Running group; ComR: Combined Running group. BV/TV: Bone Volume/Tissue Volume; Tb.N: Trabecular Number; Tb.Sp: Trabecular Spacing; Tb.Th: Trabecular Thickness.
(PDF)

## Author Contributions

**Conceptualization:** Andy Xavier, Céline Bourzac, Morad Bensidhoum, Hugues Portier, Stéphane Pallu.

**Data curation:** Andy Xavier, Céline Bourzac, Morad Bensidhoum, Hugues Portier, Stéphane Pallu.

**Formal analysis:** Andy Xavier, Hugues Portier, Stéphane Pallu.

**Funding acquisition:** Hugues Portier, Stéphane Pallu.

**Methodology:** Andy Xavier, Céline Bourzac, Hugues Portier, Stéphane Pallu.

**Project administration:** Andy Xavier, Céline Bourzac, Hugues Portier, Stéphane Pallu.

**Resources:** Morad Bensidhoum, Hugues Portier, Stéphane Pallu.

**Supervision:** Andy Xavier, Hugues Portier, Stéphane Pallu.

**Validation:** Andy Xavier, Céline Bourzac, Morad Bensidhoum, Catherine Mura, Hugues Portier, Stéphane Pallu.

**Visualization:** Andy Xavier, Hugues Portier, Stéphane Pallu.

**Writing – original draft:** Andy Xavier, Céline Bourzac, Morad Bensidhoum, Hugues Portier, Stéphane Pallu.

**Writing – review & editing:** Andy Xavier, Céline Bourzac, Morad Bensidhoum, Hugues Portier, Stéphane Pallu.

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
