## [Decision Letter · Decision Letter 0]

21 May 2024

PONE-D-24-16196Effect of different running protocols on bone morphology and microarchitecture of the forelimbs in a male Wistar rat model.PLOS ONE

Dear Dr. XAVIER,

Thank you for submitting your manuscript to PLOS ONE. After careful consideration, we feel that it has merit but does not fully meet PLOS ONE’s publication criteria as it currently stands. Therefore, we invite you to submit a revised version of the manuscript that addresses the points raised during the review process.

We look forward to receiving your revised manuscript.

Kind regards,

Masoud Rahmati

Academic Editor

PLOS ONE

Journal Requirements:

"Gérond’if DIM Longévité & Vieillissement Région île de France"

5. Please ensure that you include a title page within your main document. You should list all authors and all affiliations as per our author instructions and clearly indicate the corresponding author.

Reviewers' comments:

Reviewer's Responses to Questions

**Comments to the Author**

1. Is the manuscript technically sound, and do the data support the conclusions?

Reviewer #1: Yes

Reviewer #2: Yes

2. Has the statistical analysis been performed appropriately and rigorously? 

Reviewer #1: Yes

Reviewer #2: Yes

3. Have the authors made all data underlying the findings in their manuscript fully available?

Reviewer #1: Yes

Reviewer #2: Yes

4. Is the manuscript presented in an intelligible fashion and written in standard English?

Reviewer #1: Yes

Reviewer #2: Yes

5. Review Comments to the Author

Reviewer #1: This manuscript investigates the effect of exercise on the microstructure of rat bone. The manuscript is well-written and the experiments are well-designed; however, I have a main concern regarding the novelty of the research. Similar findings have been reported earlier (https://doi.org/10.1038/s41598-019-49432-2). Therefore, the authors are encouraged to highlight the novelty of this manuscript compared to the previously available literature. Specific comments to improve the manuscript are provided below.

• Line 167: Is a resolution of 4.91µm sufficient to measure cortical porosity? Since cortical porosity includes vascular canals, osteocyte lacunae, canaliculi, and nanopores, could you specify what you are measuring here?

• Line 173: A beam hardening of 100% is too high.

• Table 2: The authors have reported various cortical bone morphometric parameters for the whole bone. However, reporting cortical thickness could make more sense instead of reporting diameters. Also, comparison of site-dependent variation in parameters could highlight its association with the state of stress.

• Table 4: How was cortical porosity calculated? A cortical porosity of 30-40% is too high. Authors are strongly encouraged to repeat this analysis. The following articles could be useful: (https://doi.org/10.1016/j.jmbbm.2021.104770, https://doi.org/10.1152/japplphysiol.00948.2011 )

• Figure 3: Please add a scale bar to Figure 3.

• Figure 5: The quality of the uploaded figure is poor.

• The Discussion section is rather weak, and the findings of this research need to be compared to previously published data.

Reviewer #2: Comments

Neither your results nor conclusions in the abstract indicate if the increase or decrease in the forelimbs is physiologically positive or negative and how these changes would affect daily tasks, for example. In summary, your results and conclusion are too superficial as it does not clarify to the reader if these changes are “good” or “bad” for the mammal body. The authors need to re-write in a better and clearer way.

Confuse sentences “Physical exercise is recognized for its beneficial effects on the

whole body and particularly on the musculoskeletal system (2). It is now emerging as a non

medicinal therapeutic answer for the prevention and the treatment of degenerative diseases.”

“It is recognized that muscle activity through different mechanical stimulations (such as traction

or impacts) and their secretome (including myokines and cytokines) has beneficial effects on

bone tissue” and how is physical exercise related to this information?

L32 “whose” as a connecting element is a bit confusing. There are many typos and minor grammar mistakes. Please, improve the quality of the English.

The authors present an abbreviation without giving its definition (HIIT). Please, note that first you present the entire full name and then the abbreviation inside of the parentheses and not the opposite.

The authors also lack explanation in the introduction of the differences between HIIT and continuous physical exercise training, what is essential for the idea behind the paper.

Why didn’t the authors use female? It would be nice to have sex-differences explored in this study.

The conclusion of the paper must follow the same idea that I gave for the abstract conclusion.

In the supplementary material there is some sort of error, please correct it. “Running group. The statistically significant differences (*p-value<0.05) are summarized in Error! Reference

source not found. and Error! Reference source not found.”

Figure 5 of the supplementary material could be used at the end of your discussion to summarize the results.

6. PLOS authors have the option to publish the peer review history of their article (what does this mean?). If published, this will include your full peer review and any attached files.

Reviewer #1: No

Reviewer #2: **Yes: **RICARDO AUGUSTO LEONI DE SOUSA

---

## [Decision Letter · Decision Letter 1]

5 Aug 2024

Effect of different running protocols on bone morphology and microarchitecture of the forelimbs in a male Wistar rat model

PONE-D-24-16196R1

Dear Dr. Andy XAVIER,

We’re pleased to inform you that your manuscript has been judged scientifically suitable for publication and will be formally accepted for publication once it meets all outstanding technical requirements.

Kind regards,

Masoud Rahmati

Academic Editor

PLOS ONE

Additional Editor Comments (optional):

Reviewers' comments:

Reviewer's Responses to Questions

**Comments to the Author**

1. If the authors have adequately addressed your comments raised in a previous round of review and you feel that this manuscript is now acceptable for publication, you may indicate that here to bypass the “Comments to the Author” section, enter your conflict of interest statement in the “Confidential to Editor” section, and submit your "Accept" recommendation.

Reviewer #1: All comments have been addressed

Reviewer #2: All comments have been addressed

2. Is the manuscript technically sound, and do the data support the conclusions?

Reviewer #1: Yes

Reviewer #2: Yes

3. Has the statistical analysis been performed appropriately and rigorously? 

Reviewer #1: Yes

Reviewer #2: Yes

4. Have the authors made all data underlying the findings in their manuscript fully available?

Reviewer #1: (No Response)

Reviewer #2: Yes

5. Is the manuscript presented in an intelligible fashion and written in standard English?

Reviewer #1: (No Response)

Reviewer #2: Yes

6. Review Comments to the Author

Reviewer #1: Most of my queries has been incorporated in the revised manuscript. The figure quality still seems low. Authors should consider this before submitting final version.

Reviewer #2: (No Response)

7. PLOS authors have the option to publish the peer review history of their article (what does this mean?). If published, this will include your full peer review and any attached files.

Reviewer #1: No

Reviewer #2: **Yes: **Ricardo Augusto Leoni de Sousa

---

## [Editor Report · Acceptance letter]

10 Sep 2024

PONE-D-24-16196R1 

PLOS ONE

Dear Dr. Xavier, 

I'm pleased to inform you that your manuscript has been deemed suitable for publication in PLOS ONE. Congratulations! Your manuscript is now being handed over to our production team.

Kind regards, 

on behalf of

Dr. Masoud Rahmati 

Academic Editor

PLOS ONE